# New constructions of equality test scheme for cloud-assisted wireless sensor networks

**Huijun Zhu**[1,2,3]*, **Dong Xie**[4], **Haseeb Ahmad**[5], **Hasan Naji Hasan Abdullah**[1]

**1** Nanyang Institute of Technology, Nanyang, Henan, China, **2** State Key Laboratory of Networking and Switching Technology, Beijing University of Posts and Telecommunications (BUPT), Beijing, China, **3** Graphic Image and Intelligent Processing in Henan Province, International Joint Laboratory, Nanyang Institute of Technology, Nanyang, Henan, China, **4** Anhui Normal University, Wuhu, Anhui, China, **5** Department of Computer Science, National Textile University, Faisalabad, Pakistan

* zhuhj1201@163.com

**Data Availability Statement:** All relevant data are within the manuscript.

**Funding:** This work was supported by the Science and Technology Department of Henan Province (no.212102310297), the Open Foundation of State

## Abstract

Public key encryption with equality test enables the user to determine whether two ciphertexts contain the same information without decryption. Therefore, it may serve as promising cryptographic technique for cloud-assisted wireless sensor networks (CWSNs) to maintain data privacy. In this paper, an efficient RSA with equality test algorithm is proposed. The presented scheme also handles the attackers based on their authorization ability. Precisely, the proposed scheme is proved to be one-way against chosen-ciphertext attack security and indistinguishable against chosen ciphertext attacks. Moreover, the experimental evaluations depict that the underlying scheme is efficient in terms of encryption, decryption, and equality testing. Thus, this scheme may be used as a practical solution in context of CWSNs, where the users may compare two ciphertexts without decryption.

## 1 Introduction

Recently, Internet of Things (IoT) as a new information network technology is booming. In order to achieve a Smart and Secure environment, Stergiou et al. proposed a scenario that try to combine the functions of the IoT with cloud computing and edge computing and big data [1]. With the development of IoT and the technology of cloud computing, cloud-assisted wireless sensor networks (CWSNs) are widely applied in many fields, such as agriculture, military, transportation, medical and other similar fields. Although, CWSNs have extensive applications, but there also exist challenges to be addressed such as reduction of energy consumption. Recently in 2020, Guermazi et al. proposed a method to reduce energy consumption as well as to extend the lifetime of wireless sensor network [2]. For the evaluation models, Cao et al. proposed five intelligent evaluation models and implemented their experiments on the Nearest Closer Protocol with the J-Sim simulation tool [3]. Al-Qerem et al. proposed the mechanism to reduce the communication delay significantly [4]. The proposed mechanism shall enable low-latency fog computing services of the IoT applications that are a delay sensitive. The security of data is another imperative issue. Practically, extensive amount of data is being transmitted and stored on distributed servers, where it may face several threats. Therefore, to protect

Key Laboratory of Networking and Switching Technology (Beijing University of Posts and Telecommunications) (SKLNST-2019-2-17) and the National Natural Science Foundation of China (no. 61801004).

**Competing interests:** The authors have declared that no competing interests exist.

the confidentiality of such data is particularly important [5]. At present, various cryptographic algorithms are introduced for CWSNs environment. However, private key is necessary to obtain information from the encrypted data, that reduces the availability of data. In order to enhance the availability and to realize the convenient access over the encrypted data, searchable encryption technology (SE) for ciphertext data retrieval has got the festivity. SE is divided into symmetric search encryption [6–8] and public key search encryption [9]. SE algorithms realize the search operation over encrypted data (without disclosing the user's private key). Subsequently, several searchable encryption algorithms have been proposed [10–14].

In 2016, Chen et al. proposed Dual-Server Public Key Encryption with Keyword Search (DS-PEKS), that utilized the smooth projective hash function to design the scheme [15]. In the same year, Chen et al. proposed a server-aided public key encryption encryption scheme with the keyword search that obtained the security against the offline keyword guessing attack [16]. Getting motivation from the Chen's scheme, Huang et al. developed a new searchable encryption scheme with dual server model, that offers the indistinguishability of keyword retrieval trapdoor and can resist the internal keyword guessing attack (KGA) attack [17]. To satisfy users requirements in a more secure way, a fuzzy keyword search scheme is also proposed, which improves the usability of the system by matching the keywords based on similar semantic or with the nearest possible file [18]. To overcome the key escrow problem in identity-based cryptography (IBC) and the cumbersome certificate problem in conventional public key encryption (PKE), Lu et al. presented a new certificateless encryption with keyword search (CLEKS) framework [19].

Since the proposed the searchable encryption schemes can only obtain the ciphertexts of related keywords, hence, it is impossible to know the information contained in the ciphertexts. The main limitation was due to the unavailability of the method to compare the ciphertexts. In 2010, Yang et al. proposed an public key encryption with equality test (PKEwET) to compare whether two ciphertexts contain the same information without decryption [20]. Since, Yang et al's scheme allows all users to perform an equality test of ciphertexts, researchers have further studied the authorization algorithm of equality test and the security of the scheme [21–25].

PKEwET is a promising cryptographic algorithm for CWSNs due to its practical applicability. The application scenario of PKEwET in context of CWSNs is depicted in Fig 1. Precisely, the users send data to sensor networks that further proceed the ciphertexts to cloud service through gateway for storage. For retrieval, the users send trapdoors and ciphertexts to the cloud service. After receiving trapdoors, the cloud service tests whether the received ciphertexts are consistent with the stored and returns the result to users.

To merge the functionality of ciphertext matching in RSA scheme [26–28], the construction of public key encryption with equality test based on RSA is proposed in this paper. For security enhancement, Fujisaki and Okamoto method is introduced into the proposed scheme. In general, the advantages of the proposed scheme can be summarized as follows:

- The idea of public key encryption with equality test is introduced into RSA scheme. The proposed scheme fills the gap of RSA algorithm in the context of equality test over ciphertext. The major target of this paper is to make the RSA algorithm enjoying the equality test of ciphertexts. To the best of our knowledge, this is the novel algorithm of RSA with equality test.

- A simple and efficient Fujisaki and Okamoto method is introduced to enhance the security of the proposed scheme. More precisely, a semantically secure public-key encryption scheme against passive adversaries is improved to a non-malleable public-key encryption scheme against adaptive chosen ciphertext attacks in the random oracle model.

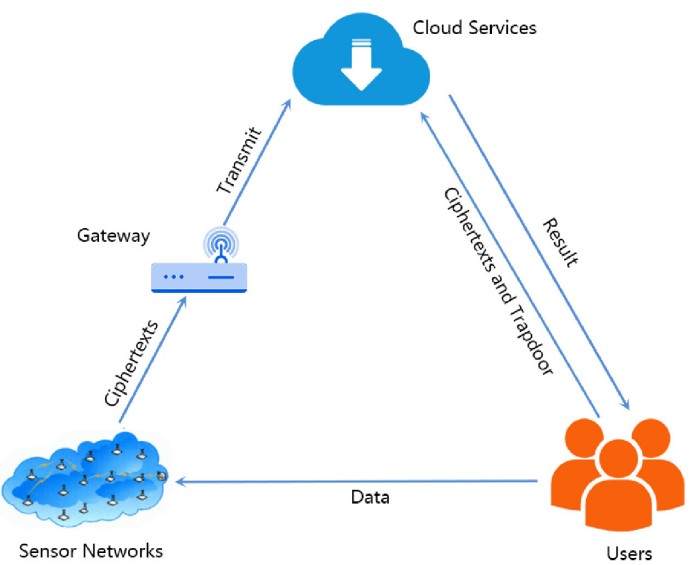

**Fig 1. The framework of the mechanism proposed.**

- To prove the efficiency of the proposed scheme, the performance analysis is presented on 512, 1024 and 2048 bits. The time of algorithm is compared in context of KenGen, encryption, decryption and test algorithms. These comparisons validate the claim of scheme efficiency.

- Since, the equality test algorithm is based on a ray. Therefore, comparing the proposal with the schemes based on bilinear pairing, it is simpler and easier to implement.

## 2 Background knowledge

### 2.1 Public-key encryption with equality test

Public-key encryption with equality test (PKEwET) allows anyone to test whether underlying ciphertexts are equal without decryption. Numerous researchers introduced authorization to the PKEwET scheme. Ma et al. proposed a primitive called PKEwET supporting flexible authorization (PKEwET-FA), which provides 4-Types flexible authorization in four different scenarios [29]. Subsequently, Zhu et al. [30] and Lin et al. [31] improved the scheme of Ma's. In order to simplify the certificate management of PKEwET, Ma et al. combined the concepts of PKEwET and identity-based encryption to present identity-based encryption with equality test (IBEwET) [32]. In 2017, Wu et al. improved Ma et al.'s scheme by reducing time computational cost [33]. Duong et al. proposed the first lattice-based PEKwET scheme in the standard model [34]. In 2020, Chen et al. introduced the equality test algorithm into blockchain and proposed blockchain-based proxy re-encryption with equality test [35].

### 2.2 Cloud-assisted wireless sensor networks

With the increasing popularity of cloud-assisted wireless sensor networks (CWSNs), the maintenance of data confidentiality has become the new challenge [36]. Therefore, several cryptographic algorithms are introduced for CWSNs environment. Wang et al. introduced homomorphic encryption into CWSNs to present a data division scheme. In the proposed

scheme, even if a forwarding node is compromised, the attacker may not be able to eavesdrop on the data [37]. In 2018, Wang et al. proposed a systematical evaluation framework for schemes to be assessed objectively [38], revisiting two foremost schemes proposed by Wu et al. [39] and Srinivas et al. [40]. The authors provided the missing evaluation for two-factor schemes in industrial CWSNs. In 2020, Li et al. proposed a multi-conditional proxy broadcast re-encryption scheme for sensor networks [41]. To offer a high level of confidentiality, Maria Azees et al. proposed an efficient affine cipher-based encryption technique [42]. Due to the decentralized nature of blockchain technology, Maria Azees et al. proposed an anonymous authentication scheme based on blockchain [43]. The proposed scheme diminishes the computational cost substantially.

In order to facilitate users to extract data from the database, Boneh et al. [44] and Bellare et al. [45] proposed searchable encryption (SE) and deterministic encryption (DE), respectively. In the application of CWSNs, SE is more useful than DE due to the searching facility is public-key encryption [46]. However, with the development of CWSNs, SE has observed some limitations, such as it is unable to perform the ciphertexts matching without decryption. To meet this challenge, we introduce PKEwET algorithm for CWSNs.

## 2.3 Notation and conventions

We denote $k$ as the security parameter. Where, $1^k$ is the string consisting of consecutive "1" bits. $A \parallel B$ is referring to the connection of strings $A$ and $B$. $\{0, 1\}^*$ is denoting the strings generated by 0 and 1. $y \leftarrow \mathcal{A}^{\mathcal{O}_1, \mathcal{O}_2, \mathcal{O}_3}(x_1, x_2, \ldots)$ denotes a probabilistic algorithm. It takes $x_1$, $x_2, \cdots$ as inputs, and outputs $y$. It allows to access the random oracles $\mathcal{O}_1, \mathcal{O}_2, \mathcal{O}_3$ before outputing the $y$. $x \in S$ indicates that $x$ is randomly selected from the set $S$.

## 2.4 Public key encryption with equality test

A PKEwET scheme is comprised of four algorithms: KeyGen, Encrypt, Decrypt and Test. The KeyGen algorithm takes a security parameter $k$ as input and outputs public key $pk$ and secret key $sk$. The Encrypt algorithm takes $pk$ and a message $M$ as inputs, and returns a ciphertext $CT$. The Decrypt algorithm takes $CT$ and $sk$ as inputs, and outputs the $M$. The Test algorithm takes the trapdoor $tr$, two ciphertexts $CT_i$ and $CT_j$ as inputs and outputs 1 or 0. As described in Fig 2, Alice uses the public key of Bob to encrypt a message $M_A$ and generates the ciphertext $CT_A$ for Bob. Bob uses the public key of Alice to encrypt a message $M_B$ and generates the ciphertext $CT_B$ for Alice. Then, the third user can perform the test algorithm and checks that whether $CT_A$ and $CT_B$ contain the same message. If $CT_A$ and $CT_B$ contain the same message, the third user outputs 1. Otherwise, the third user outputs 0.

## 2.5 Security of PKE

Here, we present two definitions of security for PKE.

**Definition 1** *One-way against chosen-ciphertext attack (OW-CCA) security: The attacker can decrypt queries at any time except for the target ciphertext $CT^*$, and the corresponding message $M$ cannot be obtained from the public key and $CT^*$.*

**Definition 2** *Indistinguishable against chosen ciphertext attacks (IND-CCA) security: The attacker can decrypt queries at any time except for the target ciphertext $CT^*$, and selects $M_0$ and $M_1$, then the challenger randomly selects $b \in \{0, 1\}$ and generates the target ciphertext $CT^*$ by $M_b$. The attacker cannot guess the value of $b$ by using ciphertext $CT^*$.*

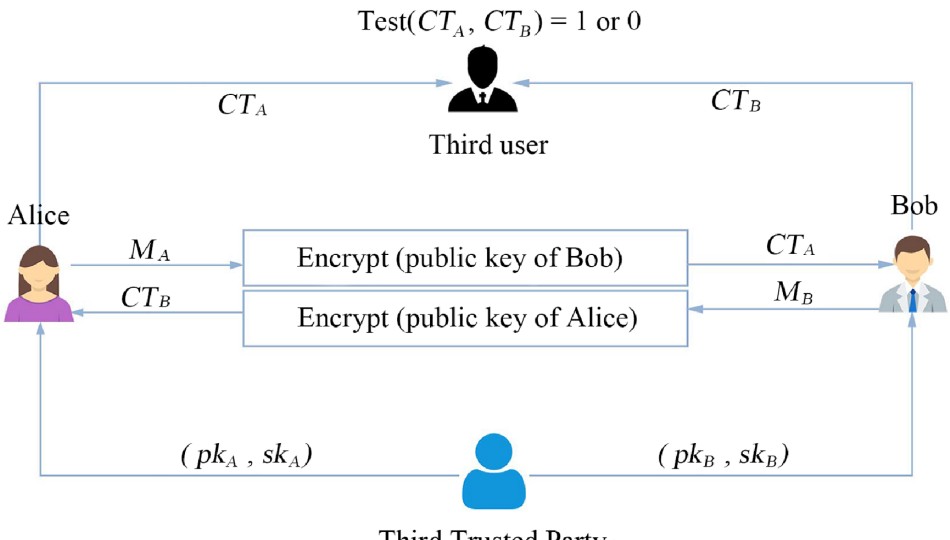

**Fig 2. The PKEwET framework [18].**

## 2.6 Organization

The rest of this paper is organized as follows: In Section 3, the security model is discussed. In Section 4, the details of the new algorithm are presented. The security of the proposed scheme is discussed in Section 5. In Section 6, the efficiency of the algorithm is evaluated by experiments. Finally, we summarize the work in Section 7.

## 3 Security models

In this section, we describe two different types of adversaries based on the adversarial permissions.

- Type-1 adversary: We authorize this adversary a trapdoor. So this type of adversary cannot recover the plaintext with the challenge ciphertext.

- Type-2 adversary: To this adversary, we do not authorize the trapdoor. So this type of adversary cannot decide that the $CT^*$ is encrypted through which message.

First, we define OW-CCA security to the Type-1 adversary in RSA with Equality Test scheme (RSA-FO-ET).

*Game 1*: Suppose that $\mathcal{A}_1$ is a Type-1 adversary and $\mathcal{S}$ is the simulator. The game between $\mathcal{A}_1$ and $\mathcal{S}$ is presented in Table 1:

**Table 1. The security model of Type-1 adversary.**

| Experiment $Exp_{\mathcal{S},\mathcal{A}_1}^{OW-CCA}$ |
| --- |
| $(pk, sk) \leftarrow KeyGen(1^k, sp)$ |
| $M \leftarrow \mathcal{A}_1^{\mathcal{O}_1,\mathcal{O}_2,\mathcal{O}_3}(i, pk)$ |
| $CT_t^* \leftarrow Enc(pk, M)$ |
| $M^* \leftarrow \mathcal{A}_1^{\mathcal{O}_4,\mathcal{O}_5,\mathcal{O}_6}(i, pk, CT_t^*)$ |
| if $M = M^*$, then return 1; |
| else return 0 |

Here, $\mathcal{O}_1(i) \triangleq KeyGen(i, sp)$, $\mathcal{O}_2(i, CT_i) \triangleq dec(sk_i, CT_i)$, $\mathcal{O}_3(i, \cdot) \triangleq ET - Auth(sk_i, \cdot)$, $\mathcal{O}_6 = \mathcal{O}_3$, but

$$\mathcal{O}_4(i) = \begin{cases} \mathcal{O}_1(i) & i \neq t \\ \bot & otherwise \end{cases}$$

$$\mathcal{O}_5(i, CT_i) = \begin{cases} \mathcal{O}_2(i, CT_i) & i \neq t, CT_i \neq CT_t^* \\ \bot & otherwise \end{cases}$$

The advantage of $\mathcal{A}_1$ in the aforementioned game is defined as follows:

$$Adv_{RSA-FO-ET,\mathcal{A}_1}^{OW-CCA}(k) = \Pr[M = M*]$$

As described in Table 1, the *KeyGen* algorithm takes $1^k$ and *sp* as inputs, then outputs the public key *pk* and private key *sk*. $\mathcal{A}_1$ asks for key queries, decryption queries and authorization queries. Then $\mathcal{S}$ initiate the challenge phase. $\mathcal{S}$ chooses a message *M* and outputs ciphertext $CT_t^*$ by performing encryption algorithm ($Enc(pk, M)$). Then, $\mathcal{A}_1$ inquires more queries in phase 2, including key queries, decryption queries and authorization queries. But it must satisfy the requirements of $\mathcal{O}_4$ and $\mathcal{O}_5$. Afterwards, $\mathcal{A}_1$ outputs a message of *M*, a guessing of $CT_t^*$.

**Definition 3** *The RSA-FO-ET scheme is OW-CCA secure if all polynomial time adversaries' advantage is negligible in the above game.*

Next, we define the IND-CCA security to the Type-2 adversary in RSA-FO-ET.

*Game 2*: Suppose that $\mathcal{A}_2$ is a Type-2 adversary and $\mathcal{S}$ is the simulator. The game between $\mathcal{A}_2$ and $\mathcal{S}$ is presented in Table 2:

Here, $\mathcal{O}_1(i) \triangleq KeyGen(i, sp)$, $\mathcal{O}_2(i, CT_i) \triangleq dec(sk_i, CT_i)$, but

$$\mathcal{O}_3(i) = \begin{cases} \mathcal{O}_1(i) & i \neq t \\ \bot & otherwise \end{cases}$$

$$\mathcal{O}_4(i, CT_i) = \begin{cases} \mathcal{O}_2(i, CT_i) & i \neq t, CT_i \neq CT^* \\ \bot & otherwise \end{cases}$$

The advantage of $\mathcal{A}_2$ in the aforementioned game is defined as follows:

$$Adv_{RSA-FO-ET,\mathcal{A}_2}^{IND-CCA}(k) = |\Pr[b = b*] - 1/2|$$

As mentioned in Table 2, the *KeyGen* algorithm takes $1^k$ and *sp* as inputs, then outputs the public key *pk* and private key *sk*. $\mathcal{A}_2$ asks for key queries and decryption queries. $\mathcal{A}_2$ chooses

**Table 2. The security model of Type-2 adversary.**

Experiment $Exp_{\mathcal{S},\mathcal{A}_2}^{IND-CCA}$
$(pk, sk) \leftarrow KeyGen(1^k, sp)$
$M_0, M_1 \leftarrow \mathcal{A}_2^{\mathcal{O}_1, \mathcal{O}_2(i, pk)}$
$b \xleftarrow{R} \{0, 1\}$
$CT_t^* \leftarrow Enc(pk, M_b)(b \in 0, 1)$
$b^* \leftarrow \mathcal{A}_2^{\mathcal{O}_3, \mathcal{O}_4}(i, pk, CT^*)$
if $b = b^*$, then return 1;
else return 0

two message $M_0$ and $M_1$ for $\mathcal{S}$. Then, $\mathcal{S}$ performs challenge phase. $\mathcal{S}$ chooses a message $M_b$ ($b \xleftarrow{R} \{0, 1\}$) and outputs ciphertext $CT_t^*$ by performing encryption algorithm ($Enc(pk, M_b)$). Then, $\mathcal{A}_2$ inquires more queries in phase 2, such as key queries and decryption queries. But it must satisfy the requirements of $\mathcal{O}_4$ and $\mathcal{O}_5$. Then, $\mathcal{A}_2$ outputs the guess of $b$.

**Definition 4** *The RSA-FO-ET scheme is IND-CCA secure if all polynomial time adversaries' advantage is negligible in the above game.*

## 4 Proposed construction

In this part, we present the details of proposed PKEwET-FA-RSA scheme as follows.

1) **Setup**($k$): To generate the system public parameters $sp$, the algorithm takes a security parameter $k$ as input and executes as follows:

- Choose the hash functions: $H_1$: $\{0, 1\}^k \rightarrow \{0, 1\}^k$, $H_2$: $\{0, 1\}^{2k} \rightarrow \{0, 1\}^k$, $H_3$: $\{0, 1\}^{5k} \rightarrow \{0, 1\}^{2l}$, and $H_4$, $H_5$: $\{0, 1\}^k \rightarrow Z_q$, here $l$ expresses the length of elements in $Z_q$.

2) **KeyGen**($sp$): This algorithm generates public keys and private keys, the details are discussed as follows:

- Select four large prime numbers $p_1$, $q_1$, $p_2$ and $q_2$ randomly and keep them confidential.;

- Compute $N_1 = p_1 {}^* q_1$ and $\varphi(N_1) = (p_1 - 1){}^*(q_1 - 1)$. Here $\varphi(N_1)$ is the Euler function value of $N_1$;

- Compute $N_2 = p_2 {}^* q_2$ and $\varphi(N_2) = (p_2 - 1){}^*(q_2 - 1)$. Here $\varphi(N_2)$ is the Euler function value of $N_2$;

- Select integers $e_1$, $e_2$ randomly. Here $e_1$, $e_2$ satisfy the following conditions:

  - $1 < e_1 < \varphi(N_1)$, $1 < e_2 < \varphi(N_2)$;

  - $gcd(e_1, \varphi(N_1)) = 1$ and $gcd(e_2, \varphi(N_2)) = 1$.

- Compute $d_1$ and $d_2$. Here $d_1$ and $d_2$ satisfy the conditions $d_1 {}^* e_1 = 1 \bmod \varphi(N_1)$ and $d_2 {}^* e_2 = 1 \bmod \varphi(N_2)$, respectively;

- Output the public key $pk = (e_1, e_2, N_1, N_2)$ and secret key $sk = (d_1, d_2)$.

3) **Encrypt**($M$, $pk$): This algorithm outputs the ciphertext $CT = (C_1, C_2, C_3, C_4, C_5)$ as follows:

- Generate a ray.

  - Use $H_4$, $H_5$ to generate a point $p = (H_4(M), H_5(M))$;

  - Construct a ray $f(x)$ using the point $p$ and the origin;

- To generate a point on the line randomly, choose $x_1 \in \{0, 1\}^l$ randomly and compute $f(x_1) = y_1$. If $x_1 = 0$, then, take $x_1 \in \{0, 1\}^l$ randomly again.

- Choose $r_1, r_2 \in Z_q^*$ randomly. Then, output the ciphertext $CT = (C_1, C_2, C_3, C_4, C_5)$ as follows:

$$C_1 = r_1^{e_1} \tag{1}$$

$$C_2 = r_2^{e_2} \tag{2}$$

$$C_3 = M||r_2 \oplus H_1(r_1) \tag{3}$$

$$C_4 = H_2(M, r_1) \tag{4}$$

$$C_5 = x_1||y_1 \oplus H_3(r_2, C_1, C_2, C_3, C_4) \tag{5}$$

4) **Decrypt**($CT$, $sk$): On input $sk$ and a ciphertext $CT = (C_1, C_2, C_3, C_4, C_5)$, the algorithm computes as follows:

$$r_1 = C_1^{d_1} \tag{6}$$

$$M||r_2 = C_3 \oplus H_1(r_1) \tag{7}$$

$$x_1||y_1 = C_5 \oplus H_3(r_2, C_1, C_2, C_3, C_4) \tag{8}$$

Use $M$ to generate $f(x)$ as in algorithm **Encrypt**. Then, check whether $C_2 = r_2^{e_2}$, $C_4 = H_2(M, r_1)$, $C_5 = x_1||y_1 \oplus H_3(r_2, C_1, C_2, C_3, C_4)$ and $f(x_1) = y_1$ all hold. If yes, the algorithm outputs $M$; otherwise, it outputs $\perp$.

5) **The authorization and test algorithm**:

Suppose $u_i$ and $u_j$ are two users in the system and $CT_i = (C_{i,1}, C_{i,2}, C_{i,3}, C_{i,4}, C_{i,5})$ (resp. $CT_j = (C_{j,1}, C_{j,2}, C_{j,3}, C_{j,4}, C_{j,5})$) is a ciphertext of $u_i$ (resp. $u_j$). $r_{2,i} \in Z_q^*$ (resp. $r_{2,j} \in Z_q^*$) denotes a randomness used in the generation of $CT_i$ (resp. $CT_j$).

- The authorization algorithm (Auth) performs as follows:
  For user $u_i$, the authorized private key is $d_{i,2}$ and the trapdoor is $td_i = d_{i,2}$;
  For user $u_j$, the authorized private key is $d_{j,2}$ and the trapdoor is $td_j = d_{j,2}$;

- The test algorithm (Test) performs as follows:
  This algorithm takes inputs: $td_i$, $td_j$ and $CT_i$, $CT_j$ and computes as follows:

$$r_{i,2} = C_{i,2}^{d_{i,2}} \tag{9}$$

$$r_{j,2} = C_{j,2}^{d_{j,2}} \tag{10}$$

$$x_i||y_i = C_{i,5} \oplus H_3(r_{i,2}, C_{i,1}, C_{i,2}, C_{i,3}, C_{i,4}) \tag{11}$$

$$x_j||y_j = C_{j,5} \oplus H_3(r_{j,2}, C_{j,1}, C_{j,2}, C_{j,3}, C_{j,4}) \tag{12}$$

Use $x_i$, $y_i$ and $x_j$, $y_j$ to check

$$x_i^{-1} * y_i = x_j^{-1} * y_j \tag{13}$$

whether holds.
Output 1 if established, otherwise output 0.

## 5 Security analysis

This section analyzes the security of the proposed scheme and authorization.

**Table 3. Phase1 of Game 1.**

| $\mathcal{S}$ | Public Channel | $\mathcal{A}_1$ | Queries |
|---|---|---|---|
| If the query $\alpha_i$ already exists in the $H_1$ list in the form $(\alpha_i, \beta_i)$ | $\xleftarrow{\alpha_i}$ | Chooses $\alpha_i$ | $H_1$-query |
| | $\xrightarrow{\beta_i}$ | Gets $\beta_i$ | |
| Otherwise $\mathcal{S}$ chooses $\beta_i \in \{0, 1\}^k$ randomly and adds the tuple $(\alpha_i, \beta_i)$ to the $H_1$ list | $\xrightarrow{\beta_i}$ | Gets $\beta_i$ | |
| If the query $(\eta_i, \theta_i)$ already exists in the $H_2$ list in the form $(\eta_i, \theta_i, \vartheta_i)$ | $\xleftarrow{\eta_i, \theta_i}$ | Chooses $(\eta_i, \theta_i)$ | $H_2$-query |
| | $\xrightarrow{\vartheta_i}$ | Gets $\vartheta_i$ | |
| Otherwise $\mathcal{S}$ chooses $\vartheta_i \in \{0, 1\}^k$ randomly and adds the tuple $(\eta_i, \theta_i, \vartheta_i)$ to the $H_2$ list | $\xrightarrow{\vartheta_i}$ | Gets $\vartheta_i$ | |
| If the query $(\delta_i, \epsilon_i, \varepsilon_i, \zeta_i, \nu_i, \xi_i)$ already exists in the $H_3$ list in the form $(\delta_i, \epsilon_i, \varepsilon_i, \zeta_i, \nu_i, \xi_i)$ | $\xleftarrow{\delta_i, \epsilon_i, \varepsilon_i, \zeta_i, \nu_i}$ | Chooses $(\delta_i, \epsilon_i, \varepsilon_i, \zeta_i, \nu_i)$ | $H_3$-query |
| | $\xrightarrow{\xi_i}$ | Gets $\xi_i$ | |
| Otherwise $\mathcal{S}$ chooses $\xi_i \in \{0, 1\}^{2l}$ randomly and adds the tuple $(\delta_i, \epsilon_i, \varepsilon_i, \zeta_i, \nu_i, \xi_i)$ to the $H_2$ list | $\xrightarrow{\xi_i}$ | Gets $\xi_i$ | |
| **KeyGen**$(pp, i)$ | $\xleftarrow{i}$ | Chooses $i (i \neq t)$ | Decryption key queries |
| | $\xrightarrow{sk_i}$ | | |
| Looks at $H_1, H_2$ and $H_3$ lists $\mathcal{S}$ runs algorithm | $\xleftarrow{CT}$ | Chooses $CT$ | Decryption queries |
| **Decrypt**$(CT, sk_i)$ | $\xrightarrow{M_i \, or \, \perp}$ | Gets $M_i$ or $\perp$ | |
| **KeyGen**$(pp, i)$ | $\xleftarrow{i}$ | Chooses $i$ | Authorization queries |
| | $\xrightarrow{sk_{i,2}}$ | Gets $sk_{i,2}$ | |

**Theorem 1** *Under the random oracle model, the proposed RSA-FO-ET scheme is OW-CCA secure against Type-1 adversary.*

Let $\mathcal{A}_1$ be a Type-1 adversary breaking the RSA-FO-ET scheme in polynomial-time. We provide $CT^* = (C_1^*, C_2^*, C_3^*, C_4^*, C_5^*)$ to the simulator $\mathcal{S}$. The aim of $\mathcal{S}$ is to recover the plaintext of $CT^*$ with a non-negligible advantage. Here, $CT^*$ is the challenge ciphertext that is generated by the challenging algorithm.

The game between $\mathcal{A}_1$ and $\mathcal{S}$ is described as follows:

First $\mathcal{A}_1$ chooses $t$ as his target at the beginning of the game. During the game, $\mathcal{S}$ maintains three watch lists of $H_1$, $H_2$ and $H_3$ and responds to $\mathcal{A}_1$ for all queries.

- 1) **Setup($1^k$)**: This algorithm takes a security parameter $1^k$ as input and outputs the system public parameters $sp = (H_1, H_2, H_3)$. Then, $\mathcal{S}$ calls the *KeyGen* algorithm, generates public/ private key pairs $(pk, sk)$, and provides the public key to $\mathcal{A}_1$.

- 2) **Phase1**: $\mathcal{A}_1$ inquires $H_1$, $H_2$ and $H_3$ queries, decryption queries and authorization queries as he/she wants. The $H_1$, $H_2$ and $H_3$ lists are initially empty. Then, $\mathcal{S}$ outputs the results accurately. Detailed descriptions are as follows:
  $\mathcal{S}$ maintains a list of 2-tuples $(\alpha_i, \beta_i)$ in $H_1$, a list of 3-tuples $(\theta_i, \vartheta_i, \eta_i)$ in $H_2$ and a list of 6-tuples $(\delta_i, \epsilon_i, \varepsilon_i, \zeta_i, \nu_i, \xi_i)$ in $H_3$. Detailed maintenances are shown in Table 3.

- 3) **Challenge**: $\mathcal{S}$ chooses $M \subset \mathcal{M}$, $r_{t,1}^* \in Z_q^*$ and $r_{t,2}^* \in Z_q^*$. Then outputs $CT^*$ as shown in Table 4.

- 4) **Phase2**: $\mathcal{A}_1$ inquires more queries as in Phase 1. But there is a condition:
  During decryption queries process, the ciphertext of $t$ is not allowed to be queried.

- 5) **Guess**: $\mathcal{A}_1$ outputs a message $M^* \subset \mathcal{M}$. If $M^* = M$, it means that $\mathcal{S}$ wins the game. Otherwise, it fails.

**Theorem 2** *Under the random oracle model, the proposed RSA-FO-ET scheme is IND-CCA secure against Type-2 adversary.*

**Table 4. Challenge.**

| $\mathcal{S}$ | Public Channel | $\mathcal{A}_1$ |
|---|---|---|
| Chooses $M$ and computes $C_1^* = r_{t,1}^{*e_1}$ $C_2^* = r_{t,2}^{*e_2}$ $C_3^* = M\|r_2 \oplus H_1(r_{t,1}^*)$ $C_4^* = H_2(M, r_{t,1}^*)$ $C_5^* = x_{t,1}\|y_{t,1}$ $\oplus H_3(r_{t,2}^*, C_1^*, C_2^*, C_3^*, C_4^*)$ | $\xrightarrow{\quad CT=(C_1^*,C_2^*,C_3^*,C_4^*,C_5^*) \quad}$ | Gets $CT$ <br><br> Inquires more queries as phase 2 |

Let $\mathcal{A}_2$ be Type-2 adversary breaking the RSA-FO-ET scheme in polynomial-time. We provide $CT^* = (C_1^*, C_2^*, C_3^*, C_4^*, C_5^*)$ to the simulator $\mathcal{S}$. The aim of $\mathcal{S}$ is to decide the plaintext of $CT^*$ is encrypted by $M_0$ or $M_1$ with a non-negligible advantage. Here, $CT^*$ is the challenge ciphertext that is generated by the challenging algorithm.

The game between $\mathcal{A}_2$ and $\mathcal{S}$ is described as follows:

First $\mathcal{A}_2$ chooses $t$ as a target at the beginning of the game. During the game, $\mathcal{S}$ maintains three watch lists of $H_1$, $H_2$ and $H_3$, then, responds $\mathcal{A}_2$ to all queries.

- 1) **Setup($1^k$)**: This algorithm takes a security parameter $1^k$ as input and outputs the system public parameters $sp = (H_1, H_2, H_3)$. Then, $\mathcal{S}$ calls the *KeyGen* algorithm and to generate public/private key pairs $(pk, sk)$, and provides the public key to $\mathcal{A}_2$.

- 2) **Phase1**: $\mathcal{A}_2$ inquires $H_1$, $H_2$ and $H_3$ queries, decryption queries and authorization queries as he/she wants. The $H_1$, $H_2$ and $H_3$ lists are initially empty. Then, $\mathcal{S}$ outputs the results accurately. Detailed descriptions are as follows:
  $\mathcal{S}$ maintains a list of 2-tuples $(\alpha_i, \beta_i)$ in $H_1$, a list of 3-tuples $(\theta_i, \vartheta_i, \eta_i)$ in $H_2$ and a list of 6-tuples $(\delta_i, \epsilon_i, \varepsilon_i, \zeta_i, v_i, \xi_i)$ in $H_3$. Detailed maintenances are shown in Table 5.

- 3) **Challenge**: $\mathcal{A}_2$ submits two equal-length messages $M_0, M_1 \subset \mathcal{M}$. $\mathcal{S}$ chooses $b \in \{0, 1\}$, $r_{t,1}^* \in Z_q^*$ and $r_{t,2}^* \in Z_q^*$. Then outputs $CT^*$ as shown in Table 6.

**Table 5. Phase 1 of Game 2.**

| $\mathcal{S}$ | Public Channel | $\mathcal{A}_1$ | Queries |
|---|---|---|---|
| If the query $\alpha_i$ already exists in the $H_1$ list in the form $(\alpha_i, \beta_i)$ | $\xleftarrow{z_i}$ | Chooses $\alpha_i$ | $H_1$-query |
| | $\xrightarrow{\beta_i}$ | Gets $\beta_i$ | |
| Otherwise $\mathcal{S}$ chooses $\beta_i \in \{0, 1\}^k$ randomly and adds the tuple $(\alpha_i, \beta_i)$ to the $H_1$ list | $\xrightarrow{\beta_i}$ | Gets $\beta_i$ | |
| If the query $(\eta_i, \theta_i)$ already exists in the $H_2$ list in the form $(\eta_i, \theta_i, \vartheta_i)$ | $\xleftarrow{\eta_i, \theta_i}$ | Chooses $(\eta_i, \theta_i)$ | $H_2$-query |
| | $\xrightarrow{\vartheta_i}$ | Gets $\vartheta_i$ | |
| Otherwise $\mathcal{S}$ chooses $\vartheta_i \in \{0, 1\}^k$ randomly and adds the tuple $(\eta_i, \theta_i, \vartheta_i)$ to the $H_2$ list | $\xrightarrow{\vartheta_i}$ | Gets $\vartheta_i$ | |
| If the query $(\delta_i, \epsilon_i, \varepsilon_i, \zeta_i, v_i, \xi_i)$ already exists in the $H_3$ list in the form $(\delta_i, \epsilon_i, \varepsilon_i, \zeta_i, v_i, \xi_i)$ | $\xleftarrow{\delta_i, \epsilon_i, \varepsilon_i, \zeta_i, v_i}$ | Chooses $(\delta_i, \epsilon_i, \varepsilon_i, \zeta_i, v_i)$ | $H_3$-query |
| | $\xrightarrow{\xi_i}$ | Gets $\xi_i$ | |
| Otherwise $\mathcal{S}$ chooses $\xi_i \in \{0, 1\}^{2l}$ randomly and adds the tuple $(\delta_i, \epsilon_i, \varepsilon_i, \zeta_i, v_i, \xi_i)$ to the $H_2$ list | $\xrightarrow{\xi_i}$ | Gets $\xi_i$ | |
| **KeyGen($pp, i$)** | $\xleftarrow{i}$ | Chooses $i (i \neq t)$ | Decryption key queries |
| | $\xrightarrow{sk_i}$ | | |
| Looks at $H_1$, $H_2$ and $H_3$ lists $\mathcal{S}$ runs algorithm | $\xleftarrow{CT}$ | Chooses $CT$ | Decryption queries |
| **Decrypt($CT, sk_i$)** | $\xrightarrow{M_i \text{ or } \perp}$ | Gets $M_i$ or $\perp$ | |
| **KeyGen($pp, i$)** | $\xleftarrow{i}$ | Chooses $i$ | Authorization queries |
| | $\xrightarrow{sk_{i,2}}$ | Gets $sk_{i,2}$ | |

**Table 6. Challenge.**

| $\mathcal{S}$ | Public Channel | $\mathcal{A}_2$ |
|---|---|---|
| Chooses $b \in \{0, 1\}$ and computes $\quad C_1^* = r_{t,1}^{*e_1}$ $\quad C_2^* = r_{t,2}^{*e_2}$ $\quad C_3^* = M_b \| r_2 \oplus H_1(r_{t,1}^*)$ $\quad C_4^* = H_2(M_b, r_{t,1}^*)$ $\quad C_5^* = x_{t,1} \| y_{t,1}$ $\quad \oplus H_3(r_{t,2}^*, C_1^*, C_2^*, C_3^*, C_4^*)$ | $\xleftarrow{M_0, M_1}$ $\xrightarrow{CT^* = (C_1^*, C_2^*, C_3^*, C_4^*, C_5^*)}$ | Chooses $M_0, M_1$ Gets $CT^*$ Inquires more queries as phase 2 as phase 2 |

**Table 7. The comparison with other schemes.**

| Scheme | Confidentiality | Encrypted Matching | Authorized | Ciphertext Test |
|---|---|---|---|---|
| [26] | √ | √ | - | - |
| [27] | √ | √ | - | - |
| [28] | √ | √ | - | - |
| Our scheme | √ | √ | √ | √ |

- 4) **Phase2**: $\mathcal{A}_2$ inquires more queries as in Phase 1. But there are two conditions: During decryption queries process, the ciphertext of $t$ is not allowed to be queried. During Authorization queries process, the authorization of user $t$ is not allowed to be queried.

- 5) **Guess**: $\mathcal{A}_2$ outputs a guess $b^* \in \{0, 1\}$. If $b^* = b$, it means that $\mathcal{S}$ wins the game. Otherwise, it fails.

## 6 Performance analysis

In this section, we discuss the performance of the proposed scheme. More precisely, the safety parameters of 512 bits, 1024 bits and 2048 bits are tested as depicted in Tables 8–10. The prototype system is developed using go 1.14.4. It is executed on intel(R) Core(TM) i7-6700 CPU 2.6GHz, 16.00GB of RAM. Fig 3 describes the performance of the proposed scheme in terms of the KenGen, Encrypt, Decrypt and Equality test algorithms. As shown in Fig 3, we

**Table 8. Computational efficiency of 512bit.**

| Algorithm | runtime(ms) |
|---|---|
| KenGen | 0.39 |
| Encrypt | 0.64 |
| Decrypt | 0.58 |
| Test | 0.12 |

**Table 9. Computational efficiency of 1024bit.**

| Algorithm | runtime(ms) |
|---|---|
| KenGen | 3.5 |
| Encrypt | 4.9 |
| Decrypt | 4.9 |
| Test | 23 |

Table 10. Computational efficiency of 2048bit.

| Algorithm | runtime(s) |
|---|---|
| KenGen | 1.6 |
| Encrypt | 3.7 |
| Decrypt | 4.6 |
| Test | 7 |

performed the experiment on its runtime while increasing length of the security parameter. It may be observed that the proposed scheme depicts efficiency. In Table 7, we compare the proposed scheme with other approaches. All of the previous RSA schemes realize protection of the confidentiality of data, while only the proposed scheme supports ciphertext data matching as well. In Table 8, the computational efficiency of 512 bits is depicted. In Table 9, the computational efficiency of 1024 bits is shown. The computational efficiency of 2048 bits is shown in Table 10. In Table 11, we present the comparison with the earlier PKEwET schemes while

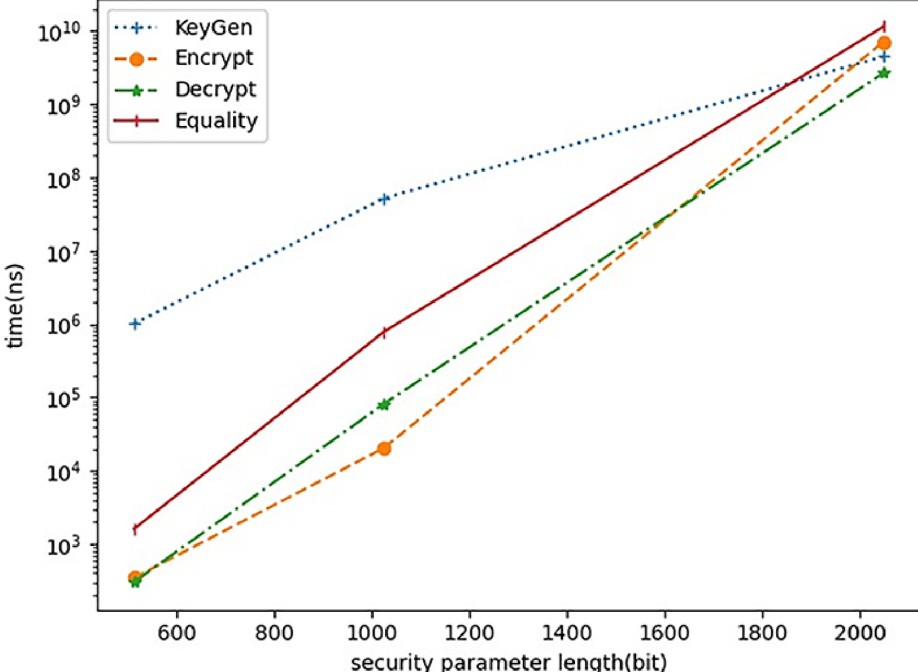

Fig 3. The performance of the proposed scheme.

Table 11. The comparison of computational complexity with others.

|  | [29] | [32] | [30] | [31] | Our scheme |
|---|---|---|---|---|---|
| $C_{Enc}$ | 6E | 6E+2P | 3E+2I | 4E+3I | 2E+1I |
| $C_{Dec}$ | 5E | 2E+2P+2I | 2E+1I | 3E+3I | 2E+1I |

$C_{Enc}$ and $C_{Dec}$: the computation complexity of algorithms forencryption and decryption; E, P and I: the exponentiation operation, the pairing operation and the inversion operation in the group $\mathbb{G}$.

considering the computation complexity of encryption and decryption algorithms. It shows that the presented scheme achieves lower computational complexity.

## 7 Conclusions

Different from the previous equality test schemes, a noval RSA with equality test scheme is proposed in this paper. To the best of our knowledge, it is the first attempt to integrate the equality test algorithm into RSA scheme. We introduced two types of attackers based on their privileges. The proposed scheme is proved to be one-way against chosen-ciphertext attack secure and indistinguishable against chosen ciphertext attacks secure. Moreover, we applied experiments on KeyGen, Encrypt, Decrypt and Equality algorithms to verify the rationality of proposed scheme in CWSNs scenario.

## Author Contributions

**Writing – original draft:** Huijun Zhu.

**Writing – review & editing:** Dong Xie, Haseeb Ahmad, Hasan Naji Hasan Abdullah.

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
