## [Decision Letter · Decision Letter 0]

28 Jun 2021

PONE-D-21-14341

New Constructions of Equality Test Scheme for Cloud-Assisted Wireless Sensor Networks

PLOS ONE

Dear Dr. Zhu,

Thank you for submitting your manuscript to PLOS ONE. After careful consideration, we feel that it has merit but does not fully meet PLOS ONE’s publication criteria as it currently stands. Therefore, we invite you to submit a revised version of the manuscript that addresses the points raised during the review process.

We look forward to receiving your revised manuscript.

Kind regards,

Pandi Vijayakumar, Ph.D

Academic Editor

PLOS ONE

Journal Requirements:

"This work was supported by the Projects of Henan Provincial Department of Science

and Technology (no.212102310297), the Open Foundation of State Key Laboratory of

Networking and Switching Technology (Beijing University of Posts and

Telecommunications) (SKLNST-2019-2-17) and the National Natural Science

Foundation of China (no. 61801004)."

3. We note you have included a table to which you do not refer in the text of your manuscript. Please ensure that you refer to Tables 8-10 in your text; if accepted, production will need this reference to link the reader to the Table.

Reviewers' comments:

Reviewer's Responses to Questions

**Comments to the Author**

1. Is the manuscript technically sound, and do the data support the conclusions?

Reviewer #1: Partly

Reviewer #2: Yes

2. Has the statistical analysis been performed appropriately and rigorously? 

Reviewer #1: Yes

Reviewer #2: Yes

3. Have the authors made all data underlying the findings in their manuscript fully available?

Reviewer #1: Yes

Reviewer #2: Yes

4. Is the manuscript presented in an intelligible fashion and written in standard English?

Reviewer #1: Yes

Reviewer #2: Yes

5. Review Comments to the Author

Reviewer #1: 1. The introduction section should be rewritten in a better way to enhance the readability.

2. The authors are requested to analyze the security strength of the proposed work with the following papers.

An efficient anonymous authentication and confidentiality preservation schemes for secure communications in

wireless body area networks M Azees, P Vijayakumar, M Karuppiah, A Nayyar

Wireless Networks 27 (3), 2119-2130.

BBAAS: Blockchain-Based Anonymous Authentication Scheme for Providing Secure Communication in VANETs

A Maria, V Pandi, JD Lazarus, M Karuppiah, MS Christo

Security and Communication Networks 2021.

3. The performance analysis part is not sufficient, more results should be added related with equality test.

Reviewer #2: New Constructions of Equality Test Scheme for Cloud-Assisted Wireless Sensor Networks is presented in this paper. It is an interesting topic and contribution is also good.  However, I have following queries on this paper:

Explain the novelty of your work presented in this work.

Paper needs to polish and provide a detailed explanation of theoretical aspects such as conditions and theorems, and practical issues like algorithms, rules and possible applications.

The Introduction section needs to be re-written to improve its quality and readability.

Improve the quality of illustrations in this paper and explain those properly.

Following are some of relevant and recent references which need to be discussed in the revised manuscript:

Sensored Semantic Annotation for Traffic Control Based on Knowledge Inference in Video

A Multi-Conditional Proxy Broadcast Re-Encryption Scheme for Sensor Networks

Modeling deep learning neural networks with denotational mathematics in UbiHealth environment

Evaluation models for the nearest closer routing protocol in wireless sensor networks

A novel energy consumption approach to extend the lifetime for wireless sensor network

IoT-based Big Data secure management in the Fog over a 6G Wireless Network

IoT transaction processing through cooperative concurrency control on fog-cloud computing environment

Many references are with incomplete bibliographic information. This must be corrected

It seems that the contribution points of the article are a little bit few. After or in the section of Motivation, it is recommended that the authors summarize the contribution points of their work, which clearly demonstrate the innovations.

There are many English and grammatical issues in the paper which need to be rectified.  

The formula character format is best to be different from the main text, and mathematical characters are recommended.

6. PLOS authors have the option to publish the peer review history of their article (what does this mean?). If published, this will include your full peer review and any attached files.

Reviewer #1: No

Reviewer #2: No

---

## [Author Response · Author response to Decision Letter 0]

6 Aug 2021

Dear Editor，

Thank you very much for your letter and also thank the reviewers for giving us constructive suggestions. Those comments are valuable and very helpful for revising and improving our paper (PONE-D-21-14341). We have studied the comments carefully and have made correction point by point. We hope that these revisions improve the paper such that you now deem it worthy of publication in PLOS ONE. Note that almost all changes are highlighted in yellow in our submitted 'Revised Manuscript with Track Changes' file. 

Thank you and best regards.

Yours sincerely,

Huijun Zhu

The following is a point-to-point response to the editor and the two reviewers’ comments. 

Review Comments to the Author

Reviewer #1: 

1.The introduction section should be rewritten in a better way to enhance the readability.

Response: 

Thank you for your careful comments. We have rewritten the Introduction thoroughly (please see Page1 and 2). 

 With the development of Internet of Things (IoT) and the technology of cloud computing [1], cloud-assisted wireless sensor networks (CWSNs) are widely applied in many fields, such as agriculture, military, transportation, medical and other similar fields. Although, CWSNs have extensive applications, but there also exist challenges to be addressed such as reduction of energy consumption. Recently in 2020, Guermazi et al. proposed a method to reduce energy consumption as well as to extend the lifetime of wireless sensor network [2]. For the evaluation models, Cao et al. proposed five intelligent evaluation models and implemented their experiments on the Nearest Closer Protocol with the J-Sim simulation tool [3]. The security of data is another imperative issue. Practically, extensive amount of data is being transmitted and stored on distributed servers, where it may face several threats. Therefore, to protect the confidentiality of such data is particularly important[4]. At present, various cryptographic algorithms are introduced for CWSNs environment. However, private key is necessary to obtain information from the encrypted data, that reduces the availability of data. …… Since, the equality test algorithm is based on a ray. Therefore, comparing the proposal with the schemes based on bilinear pairing, it is simpler and easier to implement. 

2. The authors are requested to analyze the security strength of the proposed work with the following papers.

An efficient anonymous authentication and confidentiality preservation schemes for secure communications in wireless body area networks M Azees, P Vijayakumar, M Karuppiah, A Nayyar Wireless Networks 27 (3), 2119-2130.

BBAAS: Blockchain-Based Anonymous Authentication Scheme for Providing Secure Communication in VANETs A Maria, V Pandi, JD Lazarus, M Karuppiah, MS Christo Security and Communication Networks 2021.

Response:

We agree with you. According to your valuable suggestions, we have read some works carefully.

To offer a high level of confidentiality, Maria Azees et al. proposed an efficient affine cipher-based encryption technique [37]. Due to the decentralized nature of blockchain technology, Maria Azees et al. proposed an anonymous authentication scheme based on blockchain [38]. The proposed scheme diminishes the computational cost substantially. 

3. The performance analysis part is not sufficient, more results should be added related with equality test.

Thank you sincerely for your comments on our work. According to your valuable suggestions, we have read some works about the schemes with equality test carefully. We have described the performance analysis form encryption and decryption algorithms in references of [23-26] to our works. Details are as follows:

Table 11. The comparison of computational complexity with others

 [23] [24] [25] [26] Our scheme

CEnc 4E+3I 6E 6E+2P 3E+2I 2E+1I

CDec 3E+3I 5E 2E+2P+2I 2E+1I 2E+1I

CEnc and CDec: the computation complexity of algorithms for encryption and decryption; E, P and I: the exponentiation operation, the pairing operation and the inversion operation in the group G.

In Table 11, we present the comparison with the earlier PKEwET schemes while considering the computation complexity of encryption and decryption algorithms. It shows that the presented scheme achieves lower computational complexity. 

Again, we sincerely appreciate the time you spent in reviewing our paper.

Reviewer #2: New Constructions of Equality Test Scheme for Cloud-Assisted Wireless Sensor Networks is presented in this paper. It is an interesting topic and contribution is also good.  However, I have following queries on this paper:

1.Explain the novelty of your work presented in this work.

Response: 

Thank you sincerely for your comments on our work. We have explain the novelty of our work as follows:

 In our work, we introduce the idea of public key encryption with equality test into RSA scheme. The proposed scheme fills the gap of RSA algorithm in the context of equality test over ciphertext. To the best of our knowledge, this is the novel algorithm of RSA with equality test. To enhance the security of the scheme, a simple and efficient Fujisaki and Okamoto method is introduced. More precisely, To prove the efficiency of the proposed scheme, the performance analysis is presented on 512, 1024 and 2048 bits. The time of algorithm is compared in context of KenGen , encryption, decryption and test algorithms. These comparisons validate the claim of scheme efficiency. To simplify the scheme, the equality test algorithm is based on a ray. Therefore, comparing the proposal with the schemes based on bilinear pairing, it is simpler and easier to implement.

2.Paper needs to polish and provide a detailed explanation of theoretical aspects such as conditions and theorems, and practical issues like algorithms, rules and possible applications.

Response: 

Thanks for your friendly comments. We have introduced security algorithm of the Table 1 and Table 2 into our work as follows: 

As described in Table 1, the KeyGen algorithm takes 1k and sp as inputs, then, outputs the public key pk and private key sk. A1 asks for key queries, decryption queries and authorization queries. Then S initiate the challenge phase. S chooses a message m and outputs ciphertext ct* by performing encryption algorithm Enc(pk,m). Then, A1 inquires more queries in phase 2, including key queries, decryption queries and authorization queries. But it must satisfy the requirements of O4 and O5. Afterwards, A1 outputs a message of m, a guessing of ct*.

As mentioned in Table 2, the KeyGen algorithm takes 1k and sp as inputs, then, outputs the public key pk and private key sk. A2 asks for key queries and decryption queries. A2 chooses two message m0 and m1 for S. Then, S performs challenge phase. S chooses a message mb

 (b {0,1}) and outputs ciphertext ct* by performing encryption algorithm (Enc(pk,mb)). Then, A2 inquires more queries in phase 2, such as key queries and decryption queries. But it must satisfy the requirements of O4 and O5. Then, A2 outputs the guess of b.

3.The Introduction section needs to be re-written to improve its quality and readability.

Improve the quality of illustrations in this paper and explain those properly.

Following are some of relevant and recent references which need to be discussed in the revised manuscript:

Sensored Semantic Annotation for Traffic Control Based on Knowledge Inference in Video

A Multi-Conditional Proxy Broadcast Re-Encryption Scheme for Sensor Networks

Modeling deep learning neural networks with denotational mathematics in UbiHealth environment

Evaluation models for the nearest closer routing protocol in wireless sensor networks

A novel energy consumption approach to extend the lifetime for wireless sensor network

IoT-based Big Data secure management in the Fog over a 6G Wireless Network

IoT transaction processing through cooperative concurrency control on fog-cloud computing environment

Response: 

We agree with you. Again, we sincerely appreciate the time you spent in reviewing our paper.

We have rewrited the Introduction and redrawned the picture (see Page 1-3). Moreover we have added some typical studies in Background Knowledge (see Page1 and 4). 

With the development of Internet of Things (IoT) and the technology of cloud computing [1], cloud-assisted wireless sensor networks (CWSNs) are widely applied in many fields, such as agriculture, military, transportation, medical and other similar fields. Although, CWSNs have extensive applications, but there also exist challenges to be addressed such as reduction of energy consumption. Recently in 2020, Guermazi et al. proposed a method to reduce energy consumption as well as to extend the lifetime of wireless sensor network [2]. For the evaluation models, Cao et al. proposed five intelligent evaluation models and implemented their experiments on the Nearest Closer Protocol with the J-Sim simulation tool [3].

In 2020, Li et al. proposed a multi-conditional proxy broadcast re-encryption scheme for sensor networks [41].

4.Many references are with incomplete bibliographic information. This must be corrected.

Thanks for your friendly reminder. We have revised the references carefully, such as:

“Boneh D. CrescenzoG.D, OstrovskyR, PersianoG. Public key encryption with 

keyword search. In: Cachin C., Camenisch, J.L. (eds.) EUROCRYPT. 2004, 3027, 

506-522. ” has been changed to “ Boneh D. CrescenzoG.D, OstrovskyR, PersianoG. Public key encryption with keyword search. In: Cachin C., Camenisch, J.L. (eds.) EUROCRYPT. 2004, 3027, 506-522. ”.

5.It seems that the contribution points of the article are a little bit few. After or in the section of Motivation, it is recommended that the authors summarize the contribution points of their work, which clearly demonstrate the innovations.

Thank you sincerely for your comments on our work. We have rewrited the contribution of our work as follows:

 The idea of public key encryption with equality test is introduced into RSA scheme. The proposed scheme fills the gap of RSA algorithm in the context of equality test over ciphertext. The major target of this paper is to make the RSA algorithm enjoying the equality test of ciphertexts. To the best of our knowledge, this is the novel algorithm of RSA with equality test.

A simple and efficient Fujisaki and Okamoto method is introduced to enhance the security of the proposed scheme. More precisely, a semantically secure public-key encryption scheme against passive adversaries is improved to a non-malleable public-key encryption scheme against adaptive chosen ciphertext attacks in the random oracle model.

To prove the efficiency of the proposed scheme, the performance analysis is presented on 512, 1024 and 2048 bits. The time of algorithm is compared in context of KenGen , encryption, decryption and test algorithms. These comparisons validate the claim of scheme efficiency.

 Since, the equality test algorithm is based on a ray. Therefore, comparing the proposal with the schemes based on bilinear pairing, it is simpler and easier to implement.

6. There are many English and grammatical issues in the paper which need to be rectified. 

We agree with you. We have checked the manuscript carefully again both in English and in depth to improve the quality of our manuscript. Here we list some corrections ( for an incomplete list) as follows: 

(1) “ Although the application of CWSNs is very extensive, it still faces many challenges, 

 especially the energy consumed. ” has been changed to “ Although, CWSNs have extensive applications, but there also exist challenges to be addressed such as reduction of energy consupmtion. ”

(2)“ Practically, abundant of data are transmitting that is stored on distributed servers. ” has been changed to “ extensive amount of data is being transmitted and stored on distributed servers. ” 

(3)“. These sensitive data are faced with various threats.” has been changed to “, where it may face several threats.”

(4)“ When the PKEwET scheme is applied in CWSNs, such as in Fig.1, ” has been changed to “The application scenario of PKEwET in context of CWSNs is depicted in Fig. 1.”

(5)“ To enhance the security of the scheme, the method of Fujisaki and Okamoto is introduced into the proposed scheme. ” has been changed to “For security enhancement, Fujisaki and Okamoto is introduced into the proposed scheme.”

(6)“ chosenciphertext ” has been changed to “ chosen ciphertext ”.

(7)“We can see that the proposed scheme is very efficient.” has been changed to “ These comparisons validate the claim of scheme afficiency. ”

(8)“ The time of algorithm is described from KenGen algorithm, encryption algorithm, decryption algorithm and test algorithm. ” has been changed to “ The time of algorithm is compared in context of KenGen, encryption, decryption and test algorithms. ”

Again, we sincerely appreciate the time you spent in reviewing our paper.

---

## [Decision Letter · Decision Letter 1]

25 Aug 2021

PONE-D-21-14341R1

New Constructions of Equality Test Scheme for Cloud-Assisted Wireless Sensor Networks

PLOS ONE

Dear Dr. Zhu,

Thank you for submitting your manuscript to PLOS ONE. After careful consideration, we feel that it has merit but does not fully meet PLOS ONE’s publication criteria as it currently stands. Therefore, we invite you to submit a revised version of the manuscript that addresses the points raised during the review process.

We look forward to receiving your revised manuscript.

Kind regards,

Pandi Vijayakumar, Ph.D

Academic Editor

PLOS ONE

Additional Editor Comments:

The paper needs a major revision. The authors should carefully address the reviewers comments.

Reviewers' comments:

Reviewer's Responses to Questions

**Comments to the Author**

1. If the authors have adequately addressed your comments raised in a previous round of review and you feel that this manuscript is now acceptable for publication, you may indicate that here to bypass the “Comments to the Author” section, enter your conflict of interest statement in the “Confidential to Editor” section, and submit your "Accept" recommendation.

Reviewer #1: All comments have been addressed

Reviewer #2: (No Response)

2. Is the manuscript technically sound, and do the data support the conclusions?

Reviewer #1: Yes

Reviewer #2: No

3. Has the statistical analysis been performed appropriately and rigorously? 

Reviewer #1: Yes

Reviewer #2: (No Response)

4. Have the authors made all data underlying the findings in their manuscript fully available?

Reviewer #1: Yes

Reviewer #2: No

5. Is the manuscript presented in an intelligible fashion and written in standard English?

Reviewer #1: Yes

Reviewer #2: (No Response)

6. Review Comments to the Author

Reviewer #1: (No Response)

Reviewer #2: I have gone through the revised paper. However, many of the comments are not addressed well. The quality of paper is very poor.  Conceptual background still looks poor. Related work section is still weak and several recent work needs to be mentioned as suggested earlier. Writing and formatting is weak too. I suggest authors to carefully go through each and every comment and address all those comments in the revised manuscript.

7. PLOS authors have the option to publish the peer review history of their article (what does this mean?). If published, this will include your full peer review and any attached files.

Reviewer #1: No

Reviewer #2: No

---

## [Author Response · Author response to Decision Letter 1]

25 Sep 2021

Dear Editor: 

We are submitting the revised manuscript entitled “New Constructions of Equality Test Scheme for Cloud-Assisted Wireless Sensor Networks” (PONE-D-21-14341 R1) to PLOS ONE. 

We appreciate all these review comments from reviewers. All comments and suggestions have been carefully considered. We have made careful modification on the original manuscript and listed a point-by-point response to these comments. We also made additional minor editing or changes wherever necessary to improve the quality of the manuscript. Please review the revised manuscript and see our responses to the referees. 

We believe that the quality of the manuscript has been greatly improved. Thank you for your time and consideration. 

We have a small problem to trouble you. I have made a mistake that the author(s) received specific funding for this work. Details are as follows:

This work was supported by the Science and Technology Department of Henan Province (no.212102310297), the Open Foundation of State Key Laboratory of Networking and Switching Technology (Beijing University of Posts and Telecommunications) (SKLNST-2019-2-17) and the National Natural Science Foundation of China (no. 61801004).

Again, we sincerely appreciate the time you spent in our paper. Please let me know if you have questions about the manuscript.

With best regards.

Yours sincerely,

Huijun Zhu et al.

Email: zhuhj1201@163.com

The following is a point-to-point response to the editor and the two reviewers’ comments. 

Review Comments to the Author

Reviewer #1: Thank you sincerely for your approval of our work. 

Reviewer #2: New Constructions of Equality Test Scheme for Cloud-Assisted Wireless Sensor Networks is presented in this paper. It is an interesting topic and contribution is also good.  However, I have following queries on this paper:

1.Explain the novelty of your work presented in this work.

Response: 

Thank you sincerely for your comments on our work. We have explain the novelty of our work as follows:

 In our work, we introduce the idea of public key encryption with equality test into RSA scheme. The proposed scheme fills the gap of RSA algorithm in the context of equality test over ciphertext. To the best of our knowledge, this is the novel algorithm of RSA with equality test. To enhance the security of the scheme, a simple and efficient Fujisaki and Okamoto method is introduced. More precisely, To prove the efficiency of the proposed scheme, the performance analysis is presented on 512, 1024 and 2048 bits. The time of algorithm is compared in context of KenGen , encryption, decryption and test algorithms. These comparisons validate the claim of scheme efficiency. To simplify the scheme, the equality test algorithm is based on a ray. Therefore, comparing the proposal with the schemes based on bilinear pairing, it is simpler and easier to implement.

2.Paper needs to polish and provide a detailed explanation of theoretical aspects such as conditions and theorems, and practical issues like algorithms, rules and possible applications.

Response: Thanks for your friendly comments. We added a section of the algorithms.

We have introduced security algorithm of the Table 1 and Table 2 into our work as follows: 

3.The Introduction section needs to be re-written to improve its quality and readability.

Response: Thank you sincerely for your comments on our work. We have rewritten the Introduction thoroughly (please see Page1 to 4). 

Recently, Internet of Things (IoT) as a new information network technology is booming. In order to achieve a Smart and Secure environment, Stergiou et al. proposed a scenario that try to combine the functions of the IoT with cloud computing and edge computing and big data [1]. With the development of Internet of Things (IoT) and the technology of cloud computing, cloud-assisted wireless sensor networks (CWSNs) are widely applied in many fields, such as agriculture, military, transportation, medical and other similar fields. Although, CWSNs have extensive applications, but there also exist challenges to be addressed such as reduction of energy consumption. Recently in 2020, Guermazi et al. proposed a method to reduce energy consumption as well as to extend the lifetime of wireless sensor network [2]. For the evaluation models, Cao et al. proposed five intelligent evaluation models and implemented their experiments on the Nearest Closer Protocol with the J-Sim simulation tool [3]. Al-Qerem et al. proposed the mechanism to reduce the communication delay significantly [4]. ……

4.Improve the quality of illustrations in this paper and explain those properly.

Response: Thank you sincerely for your comments on our work. To improve the quality of illustrations, we have redrawn them and revised the characters in the illustrations. Details are as follows:

PKEwET is a promising cryptographic algorithm for CWSNs due to its practical applicability. The application scenario of PKEwET in context of CWSNs is depicted in Fig 1. Precisely, the users send data to sensor networks that further proceed the ciphertexts to cloud service through gateway for storage. For retrieval, the users send trapdoors and ciphertexts to the cloud service. After receiving trapdoors, the cloud service tests whether the received ciphertexts are consistent with the stored and returns the result to users.

Fig 1. The Framework of the Mechanism Proposed

As described in Fig 2, Alice uses the public key of Bob to encrypt a message and generates the ciphertext for Bob. Bob uses the public key of Alice to encrypt a message and generates the ciphertext for Alice. Then, the third user can perform the test algorithm and checks that whether and contain the same message. If and contain the same message, the third user outputs 1. Otherwise, the third user outputs 0.

Fig 2. The PKEwET framework [18].

5.Following are some of relevant and recent references which need to be discussed in the revised manuscript:

Sensored Semantic Annotation for Traffic Control Based on Knowledge Inference in Video

A Multi-Conditional Proxy Broadcast Re-Encryption Scheme for Sensor Networks

Modeling deep learning neural networks with denotational mathematics in UbiHealth environment

Evaluation models for the nearest closer routing protocol in wireless sensor networks

A novel energy consumption approach to extend the lifetime for wireless sensor network

IoT-based Big Data secure management in the Fog over a 6G Wireless Network

IoT transaction processing through cooperative concurrency control on fog-cloud computing environment

Response: 

Thank you very much for your comments to our manuscript. We agree with you. Again, we sincerely appreciate the time you spent in reviewing our paper. We have rewrited the Introduction. Moreover we have added some typical studies in Background Knowledge (see Page1 - 4). 

Recently, Internet of Things (IoT) as a new information network technology is booming. In order to achieve a Smart and Secure environment, Stergiou et al. proposed a scenario that try to combine the functions of the IoT with cloud computing and edge computing and big data [1]. With the development of Internet of Things (IoT) and the technology of cloud computing, cloud-assisted wireless sensor networks (CWSNs) are widely applied in many fields, such as agriculture, military, transportation, medical and other similar fields. Although, CWSNs have extensive applications, but there also exist challenges to be addressed such as reduction of energy consumption. Recently in 2020, Guermazi et al. proposed a method to reduce energy consumption as well as to extend the lifetime of wireless sensor network [2]. For the evaluation models, Cao et al. proposed five intelligent evaluation models and implemented their experiments on the Nearest Closer Protocol with the J-Sim simulation tool [3]. Al-Qerem et al. proposed the mechanism to reduce the communication delay significantly [4]. The proposed mechanism shall enable low-latency fog computing services of the IoT applications that are a delay sensitive. The security of data is another imperative issue. Practically, extensive amount of data is being transmitted and stored on distributed servers, where it may face several threats. Therefore, to protect the confidentiality of such data is particularly important [5]. At present, various cryptographic algorithms are introduced for CWSNs environment. However, private key is necessary to obtain information from the encrypted data, that reduces the availability of data. In order to enhance the availability and to realize the convenient access over the encrypted data, searchable encryption technology (SE) for ciphertext data retrieval has got the festivity. SE is divided into symmetric search encryption [6-8] and public key search encryption [9]. SE algorithms realize the search operation over encrypted data (without disclosing the user's private key). Subsequently, several searchable encryption algorithms have been proposed [10-14].

……

In 2020, Li et al. proposed a multi-conditional proxy broadcast re-encryption scheme for sensor networks [44].

The references are as follows：

[1] C. L. Stergiou, K. E. Psannis and B. B. Gupta. IoT-based big data secure management in the fog over a 6G wireless network. IEEE Internet of Things Journal, 2020, 8(7): 5164-5171.

[2] A. Guermazi, H. Trabelsi, W. Jerbi. A novel energy consumption approach to extend the lifetime for wireless sensor network. International Journal of High Performance Computing and Networking, 2020, 16(2-3): 160-169.

[3] N. Cao, P. Liu, G. Li, et al. Evaluation Models for the Nearest Closer Routing Protocol in Wireless Sensor Networks. IEEE Access, 2018, 6: 77043-77054.

[4] A. Al-Qerem, M. Alauthman, A. Almomani, et al. IoT transaction processing through cooperative concurrency control on fog-cloud computing environment. Soft Computing, 2020, 24(8): 5695-5711.

[13] C. Choi, T. Wang, C. Esposito, et al. Sensored Semantic Annotation for Traffic Control Based on Knowledge Inference in Video. IEEE Sensors Journal, 2021, 21(10): 11758-11768.

[22] J. Sarivougioukas, A. Vagelatos. Modeling Deep Learning Neural Networks With Denotational Mathematics in UbiHealth Environment. International Journal of Software Science and Computational Intelligence, 2020, 12(3):14-27.

[44] P. Li, L. Zhu, B. Gupta, et al. A Multi-Conditional Proxy Broadcast Re-Encryption Scheme for Sensor Networks. Computers, Materials and Continua, 2020, 65(3):2079-2090.

6.Many references are with incomplete bibliographic information. This must be corrected.

Response: Thanks for your friendly reminder. We have revised the references carefully, such as:

(1)“A. Al-Qerem, M. Alauthman, A. Almomani, et al. IoT transaction processing through cooperative concurrency control on fog-cloud computing environment. Soft Computing, 2020, 24(2-C).” has been changed to “ A. Al-Qerem, M. Alauthman, A. Almomani, et al. IoT transaction processing through cooperative concurrency control on fog-cloud computing environment. Soft Computing, 2020, 24(8): 5697-5711 .”

(2)“A. Guermazi, H. Trabelsi, W. Jerbi. A novel energy consumption approach to extend the lifetime for wireless sensor network. International Journal of High Performance Computing and Networking, 2020, 16(2/3):160.” has been changed to “A. Guermazi, H. Trabelsi, W. Jerbi. A novel energy consumption approach to extend the lifetime for wireless sensor network. International Journal of High Performance Computing and Networking, 2020, 16(2-3): 160-169” 

(3)“N. Cao, P. Liu, G. Li, et al. Evaluation Models for the Nearest Closer Routing Protocol in Wireless Sensor Networks. IEEE Access, 2018:1-1.” has been changed to “N. Cao, P. Liu, G. Li, et al. Evaluation Models for the Nearest Closer Routing Protocol in Wireless Sensor Networks. IEEE Access, 2018, 6: 77043-77054.”

(4)“X. Peng, L. Shuai, W. Wei, et al. Dynamic Searchable Symmetric Encryption with Physical Deletion and Small Leakage. Australasian Conference on Information Security and Privacy. Springer, Cham, 2017.” has been changed to “X. Peng, L. Shuai, W. Wei, et al. Dynamic Searchable Symmetric Encryption with Physical Deletion and Small Leakage. Australasian Conference on Information Security and Privacy. Springer, Cham, 201: 207-226.”

(5)“B. Dan, G. D. Crescenzo, R. Ostrovsky, et al. Public Key Encryption with Keyword Search. Advances in Cryptology - EUROCRYPT 2004, International Conference on the Theory and Applications of Cryptographic Techniques, Interlaken, Switzerland, May 2-6, 2004, Proceedings. ” has been changed to “ D. Boneh, D. G. Crescenzo, R. Ostrovsky, et al. Public key encryption with keyword search. International conference on the theory and applications of cryptographic techniques. Springer, Berlin, Heidelberg, 2004: 506-522.”

(6)“H. Zhu, L. Wang, H. Ahmad, et al. Key-Policy Attribute-Based Encryption With Equality Test in Cloud Computing. IEEE Access, 2017, PP(99):1-1.” has been changed to “H. Zhu, L. Wang, H. Ahmad, et al. Key-Policy Attribute-Based Encryption With Equality Test in Cloud Computing. IEEE Access, 2017, 5: 20428-20439.”

(7)“Lin X J, Qu H, Zhang X. Public Key Encryption Supporting Equality Test and Flexible Authorization without Bilinear Pairings. IACR Cryptology ePrint Archive 2016; 2016: 277. URL http://eprint.iacr.org/2016/277. has been changed to “X. Lin, L. Sun, H. Qu, et al. Public key encryption supporting equality test and flexible authorization without bilinear pairings. Computer Communications, 2021, 170: 190-199.”

(8)“S. Ma, Q. Huang, M. Zhang, and B. Yang.” has been changed to “S. Ma, Q. Huang, M. Zhang and B. Yang.”

(9)“X. Chen, J. Li, J. Weng, J. Ma, and W. Lou. Verifiable computation over large database with incremental updates. IEEE Transactions on Computers. vol. 65, no. 10, pp. 3184-3195, 2016. ” has been changed to “X. Chen, J. Li, J. Weng, J. Ma, and W. Lou. Verifiable computation over large database with incremental updates. IEEE Transactions on Computers. 2015, 65(10): 3184-3195.”

(10)“X. Wang and Z. Zhang, Data division scheme based on homomorphic encryption in WSNs for health care. Journal of Medical Systems, vol.39, no. 12, 2015. ” has been changed to “X. Wang and Z. Zhang, Data division scheme based on homomorphic encryption in WSNs for health care. Journal of Medical Systems, 2015, 39(12): 1-7.”

(11) “D. Wang, W. Li, P. Wang. Measuring Two-Factor Authentication Schemes for Real-Time Data Access in Industrial Wireless Sensor Networks. IEEE Transactions on Industrial Informatics, 2018:4081-4092.” has been changed to “D. Wang, W. Li, P. Wang. Measuring Two-Factor Authentication Schemes for Real-Time Data Access in Industrial Wireless Sensor Networks. IEEE Transactions on Industrial Informatics, 2018, 14(9): 4081-4092.” 

(12) “D. Boneh, Crescenzo G. D, Ostrovsky R, and Persiano G.” has been changed to “D. Boneh, G. D. Crescenzo, R. Ostrovsky and G. Persiano.”

(13) “M. Bellare, Boldyreva A. and ONeill A.” has been changed to “M. Bellare, A. Boldyreva and A. ONeill.”

(14) “F. Wu, L. Xu, S. Kumari, and X. Li.” has been changed to “ F. Wu, L. Xu, S. Kumari and X. Li. ”

(15) “J. Srinivas, S. Mukhopadhyay, and D. Mishra. Secure and efficient user authentication scheme for multi-gateway wireless sensor networks. AdHoc Netw. 2017, vol. 54, pp. 147-169. ” has been changed to “J. Srinivas, S. Mukhopadhyay and D. Mishra. Secure and efficient user authentication scheme for multi-gateway wireless sensor networks. AdHoc Netw. 2017, 54: 147-169.”

(16) “Xu P , He S , Wang W , et al. Lightweight Searchable Public-key Encryption for Cloud-assisted Wireless Sensor Networks. IEEE Transactions on Industrial Informatics, 2017:1-1. ” has been changed to “P. Xu, S. He, W. Wang, et al. Lightweight Searchable Public-key Encryption for Cloud-assisted Wireless Sensor Networks. IEEE Transactions on Industrial Informatics, 2017, 14(8): 3712-3723.”

7.It seems that the contribution points of the article are a little bit few. After or in the section of Motivation, it is recommended that the authors summarize the contribution points of their work, which clearly demonstrate the innovations.

Response: Thank you sincerely for your comments on our work. We have rewrited the contribution of our work as follows:

 The idea of public key encryption with equality test is introduced into RSA scheme. The proposed scheme fills the gap of RSA algorithm in the context of equality test over ciphertext. The major target of this paper is to make the RSA algorithm enjoying the equality test of ciphertexts. To the best of our knowledge, this is the novel algorithm of RSA with equality test.

A simple and efficient Fujisaki and Okamoto method is introduced to enhance the security of the proposed scheme. More precisely, a semantically secure public-key encryption scheme against passive adversaries is improved to a non-malleable public-key encryption scheme against adaptive chosen ciphertext attacks in the random oracle model.

To prove the efficiency of the proposed scheme, the performance analysis is presented on 512, 1024 and 2048 bits. The time of algorithm is compared in context of KenGen , encryption, decryption and test algorithms. These comparisons validate the claim of scheme efficiency.

 Since, the equality test algorithm is based on a ray. Therefore, comparing the proposal with the schemes based on bilinear pairing, it is simpler and easier to implement.

8.There are many English and grammatical issues in the paper which need to be rectified.

Response: Thanks for your friendly reminder. We scanned grammatical mistakes throughout the entire manuscript and made corresponding corrections to improve the readability and quality of the writing. Here we list some corrections (for an incomplete list) as follows: 

(1) “ Although the application of CWSNs is very extensive, it still faces many challenges, 

 especially the energy consumed. ” has been changed to “ Although, CWSNs have extensive applications, but there also exist challenges to be addressed such as reduction of energy consupmtion. ”

(2)“ Practically, abundant of data are transmitting that is stored on distributed servers. ” has been changed to “ extensive amount of data is being transmitted and stored on distributed servers. ” 

(3)“. These sensitive data are faced with various threats.” has been changed to “, where it may face several threats.”

(4)“ When the PKEwET scheme is applied in CWSNs, such as in Fig.1, ” has been changed to “The application scenario of PKEwET in context of CWSNs is depicted in Fig. 1.”

(5)“ To enhance the security of the scheme, the method of Fujisaki and Okamoto is introduced into the proposed scheme. ” has been changed to “For security enhancement, Fujisaki and Okamoto is introduced into the proposed scheme.”

(6)“ chosenciphertext ” has been changed to “ chosen ciphertext ”.

(7)“We can see that the proposed scheme is very efficient.” has been changed to “ These comparisons validate the claim of scheme afficiency. ”

(8)“ The time of algorithm is described from KenGen algorithm, encryption algorithm, decryption algorithm and test algorithm. ” has been changed to “ The time of algorithm is compared in context of KenGen, encryption, decryption and test algorithms. ”

Again, we sincerely appreciate the time you spent in reviewing our paper.

9. The formula character format is best to be different from the main text, and mathematical characters are recommended.

 Response: Thank you sincerely for your comments on our work. As many special characters are represented by upper-case English letters in contexts, such as articles of Eurocrypt conference and International Cryptology Conference, we also utilized the upper-case English letters method in the study. 

Thanks sincerely for your comments, we have scanned the full paper carefully and revised the unqualified characters in the text. Here we list some corrections (for an incomplete list) as follows: 

(1)all of the "m" has been changed to “M”, 

(2)all of the "C" has been changed to “CT”, 

(3) all of the "C*" has been changed to “CT*”.

We believe that the quality of the manuscript has been greatly improved. Thank you for your time and consideration.

---

## [Decision Letter · Decision Letter 2]

5 Oct 2021

New Constructions of Equality Test Scheme for Cloud-Assisted Wireless Sensor Networks

PONE-D-21-14341R2

Dear Dr. Zhu,

We’re pleased to inform you that your manuscript has been judged scientifically suitable for publication and will be formally accepted for publication once it meets all outstanding technical requirements.

Kind regards,

Pandi Vijayakumar, Ph.D

Academic Editor

PLOS ONE

Additional Editor Comments (optional):

The authors have done all the corrections given by the reviewers. Hence, the paper can be accepted in the present format.

Reviewers' comments:

Reviewer's Responses to Questions

**Comments to the Author**

1. If the authors have adequately addressed your comments raised in a previous round of review and you feel that this manuscript is now acceptable for publication, you may indicate that here to bypass the “Comments to the Author” section, enter your conflict of interest statement in the “Confidential to Editor” section, and submit your "Accept" recommendation.

Reviewer #1: (No Response)

Reviewer #2: (No Response)

2. Is the manuscript technically sound, and do the data support the conclusions?

Reviewer #1: (No Response)

Reviewer #2: Yes

3. Has the statistical analysis been performed appropriately and rigorously? 

Reviewer #1: (No Response)

Reviewer #2: Yes

4. Have the authors made all data underlying the findings in their manuscript fully available?

Reviewer #1: (No Response)

Reviewer #2: Yes

5. Is the manuscript presented in an intelligible fashion and written in standard English?

Reviewer #1: (No Response)

Reviewer #2: Yes

6. Review Comments to the Author

Reviewer #1: (No Response)

Reviewer #2: New Constructions of Equality Test Scheme for Cloud-Assisted Wireless Sensor Networks is presented in this paper. Paper is revised well. It can be accepted now. There is no more feedback from my side.

7. PLOS authors have the option to publish the peer review history of their article (what does this mean?). If published, this will include your full peer review and any attached files.

Reviewer #1: No

Reviewer #2: No

---

## [Editor Report · Acceptance letter]

8 Oct 2021

PONE-D-21-14341R2 

New Constructions of Equality Test Scheme for Cloud-Assisted Wireless Sensor Networks 

Dear Dr. Zhu:

I'm pleased to inform you that your manuscript has been deemed suitable for publication in PLOS ONE. Congratulations! Your manuscript is now with our production department. 

Kind regards, 

on behalf of

Dr. Pandi Vijayakumar 

Academic Editor

PLOS ONE